# Anatomy-Guided Surface Diffusion Model for Alzheimer's Disease Normative Modeling

**Jianwei Zhang**[1,2]                  JIAZHANG@LONI.USC.EDU
**Yonggang Shi**[1,2,3]                   YSHI@LONI.USC.EDU

[1]*Stevens Neuroimaging and Informatics Institute, Keck School of Medicine, University of Southern California, Los Angeles, CA, USA*

[2]*Ming Hsieh Department of Electrical and Computer Engineering, Viterbi School of Engineering, University of Southern California, Los Angeles, CA, USA*

[3]*Alfred E. Mann Department of Biomedical Engineering, Viterbi School of Engineering, University of Southern California, Los Angeles, CA, USA*

**Editors:** Accepted for publication at MIDL 2025

## Abstract

Normative modeling has emerged as a pivotal approach for characterizing heterogeneity and individual variance in neurodegenerative diseases, notably Alzheimer's disease (AD). One of the challenges of cortical normative modeling is the anatomical structure mismatch due to folding pattern variability. Traditionally, registration is applied to address this issue and recently deep generative models are employed to generate anatomically aligned samples for analyzing disease progression; however, these models are predominantly applied to volume-based data, which often falls short in capturing intricate morphological changes on the brain cortex. As an alternative, surface-based analysis has been proven to be more sensitive in disease modeling such as AD. Yet, like volume-based data, it also suffers from the mismatch problem. To address these limitations, we propose a novel generative normative modeling framework by transferring the conditional diffusion generative model to the spherical domain. Furthermore, the proposed model generates normal feature map distributions by explicitly conditioning on individual anatomical segmentation to ensure better geometrical alignment which helps to reduce variance between subjects in normative analysis. We find that our model can generate samples that are better anatomically aligned than registered reference data and through ablation study and normative assessment experiments, the samples are able to better measure individual differences from the normal distribution and increase sensitivity in differentiating cognitively normal (CN), mild cognitive impairment (MCI), and Alzheimer's disease (AD) patients.

**Keywords:** Alzheimer's Disease, Diffusion Generative Model, Cortical Surface.

## 1. Introduction

Normative modeling has been proven to be an effective approach for modeling neurodegenerative diseases such as Alzheimer's disease (Rutherford et al., 2023). The core idea of normative modeling is defining normal distribution such that each subject can be measured against it to characterize deviation from norm. One of the major challenges of such tasks is the individual anatomical variability. Specifically, the cortical folding patterns exhibit considerable heterogeneity across individuals, thereby complicating the establishment of meaningful comparisons. Conventionally, statistical analysis techniques are applied on the anatomically registered images to attenuate effect of individual variability. However, due to

shape differences, registered images still have significant gyral/sulcal mismatch (Zhang and Shi, 2023) and statistical methods are usually limited in their abilities to capture complex nonlinear relationships. As an alternative, deep generative models have recently been introduced to address these limitations. The idea is to train a generative model that encodes how normal distribution behaves. Variational Autoencoder (VAE) (Bass et al., 2020; Ravi et al., 2022), flow-based model (Hwang et al., 2019), Generative Adversarial Network (GAN) (Bai et al., 2022) were employed to model the normal distribution on the brain MRI volume space and utilize deviation of original data from generated normal samples as disease atrophy map for analysis. Although, these previous research achieved good results on the volume data, few attempts have been made to adapt these methods for cortical surface-based data, which has been proven to more prominent at capturing detailed anatomical changes(Hutton et al., 2009; Lerch and Evans, 2005).

Traditional surface-based analysis is built upon registering brain surfaces across subjects or with a template surface(Yeo et al., 2009; Fischl et al., 2004), but this process also suffers from cortical structure mismatch. To account for this problem, previous works have attempted personalized analysis where, instead of using entire dataset, only a subset with similar anatomical structures were used for analysis(Zhang and Shi, 2023, 2021). However, these approaches suffer from limited data availability and computational complexity as high cortical variability might not be represented by the existing datasets. Therefore, generative model is a promising alternative approach to generate personalized reference sets to alleviate challenges in matching against real data.

Recently, diffusion models have emerged as an effective framework for stable and effective image generation(Nichol and Dhariwal, 2021b; Song et al., 2021; Ho et al., 2020). To leverage this advancement of generative model, in this paper, we adapt the Denoising Diffusion Probabilistic Models(DDPM) framework (Ho et al., 2020) from euclidean image domain to non-euclidean spherical domain and propose a conditional surface diffusion model that utilizes gyral sulcal segmentation masks to generate cortical surface features that are anatomically aligned. The proposed model is applied to HCP(Van Essen et al., 2012) and ADNI dataset(Mueller et al., 2005) to conduct unconditional generative task, ablation study and normative modeling on cognitively normal(CN), mild cognitive impairment (MCI) and Alzheimer's disease(AD) subjects. The results show that our model is able to generate faithful and anatomically aligned feature maps and increase the sensitivity of surface based disease analysis.

## 2. Method

Our proposed method consists of three parts: surface based diffusion model with condition, denoising network in spherical domain, and normative modeling via sampling. The overall diffusion model is shown in Fig. 1.

### 2.1. Denoising Diffusion Probabilistic Models(DDPM)

DDPM (Ho et al., 2020; Rombach et al., 2022) is an iterative generative model for modeling data distribution from samples. Given a series of observed samples $x_i$, which is drawn from the data distribution p(x), the model learns to generate new samples from p(x) through a forward and backward diffusion process. The diffusion process of DDPM is governed

by a Markov chain as in equation 1 and 2, which describe forward and backward process respectively:

$$q(x_t|x_{t-1}) = \mathcal{N}(x_t; \sqrt{1-\beta_t}x_{t-1}, \beta_t\mathcal{I}) \quad q(x_{1:t}|x_0) = \prod_{t=1}^{T} q(x_t|x_{t-1}) \tag{1}$$

$$p_\theta(x_{0:T}) = p(x_T)\prod_{t=1}^{T} p_\theta(x_t|x_{t-1}), p_\theta(x_t|x_{t-1}) = \mathcal{N}(x_{t-1}; \mu_\theta(x_t, t), \Sigma\theta(x_t, t)) \tag{2}$$

where $\mathcal{N}$ is the Gaussian distribution and q is the transition probability of the forward process. $p_\theta$, $\mu_\theta$ and $\Sigma_\theta$ are parameterized estimation from neural networks. $x_0$ is the original data and $x_t$ is the noisy data after adding t steps of noise. The T denotes the total number of steps. $\beta_t$ is from a predefined set of variance schedule $\{\beta_t \in (0,1)\}|_1^T$. The information within the data is progressively destroyed by adding independent Gaussian noise for a certain number of steps in the forward process. The backward process is then formulated as a sampling process by implementing a neural network to estimate $\mu_\theta(x_t, t), \Sigma\theta(x_t, t)$ iteratively and denoise the noisy data $x_T$ to achieve new sample generation.

$$L =_{t\sim[1,T],x0,\epsilon_t} \left[||\epsilon_t - \epsilon_\theta(\sqrt{\overline{\alpha}_t}x_0 + \sqrt{1-\overline{\alpha}_t}\epsilon_t, t)||\right] \tag{3}$$

The model is trained by optimizing a simplified Evidence Lower Bound loss(Ho et al., 2020) in equation 3. $\epsilon_t$ is the noise at time t and $\epsilon_\theta$ is the neural network. $\overline{\alpha}_t$ is $\prod_{i=1}^{T}(1-\beta_i)$. We employ the cosine beta schedule(Nichol and Dhariwal, 2021a) as the variance schedule and the velocity sample scheme in (Salimans and Ho, 2022), which we empirically find to be more stable. During training, a randomly sampled t steps of noise is applied to a feature map and the resulting noisy image is the input to the network along with the time step t in the form of a time embedding vector. The loss is computed between network output and original feature map without noise. After training, the model can be iteratively applied to random noise or noisy input data for a selected number of steps to generate new samples.

### 2.2. Anatomical and Demographic Conditioning

The original DDPM is for modeling unconditional distributions. To generate samples that are better anatomically aligned, we modified the model to take additional conditions. In our method, two types of conditions are used: demographic and anatomical conditions. The demographic conditions include sex and biological age. Both values are first passed into the network through serveral multilayer perceptrons and activation layers. The embedding vectors are then added to the time embedding (Ho et al., 2020) and passed to the network. For anatomical condition,the gyral/sulcal segmentation mask(Shi et al., 2008) is concatenated with the input feature map as input to the network. All the conditions are used during training and sampling.

### 2.3. Denoising Network in Spherical Domain

To align data in a common space, the feature maps and masks are resampled to a standard icosahedron. To transfer convolution in image domain, we adapt the convolution method from (Zhao et al., 2021) in the spherical domain, which defines convolution by the 1 ring

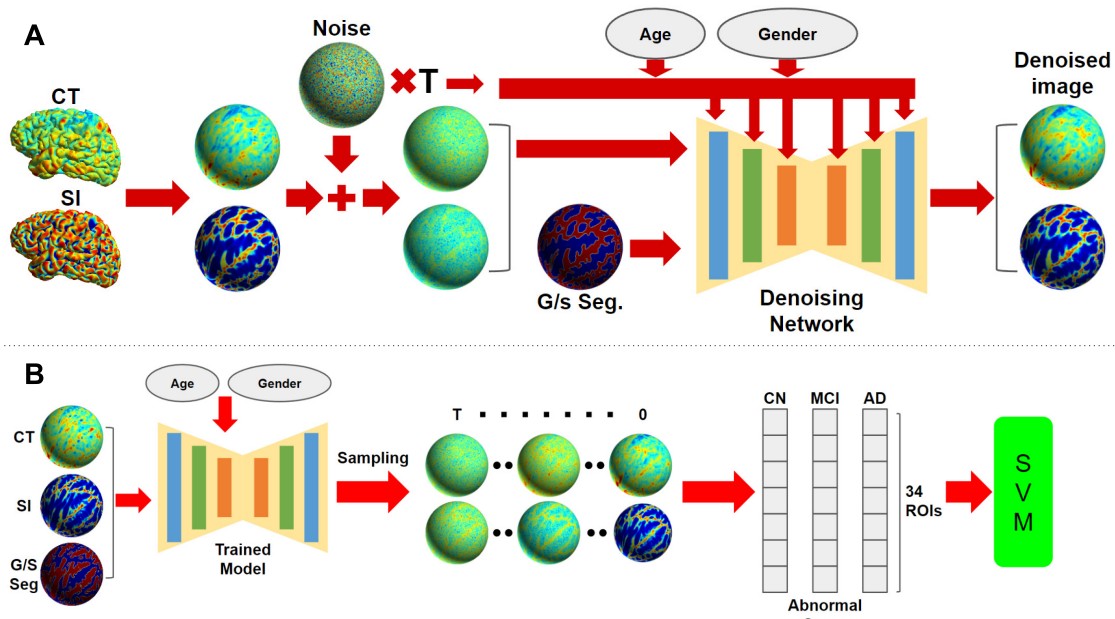

Figure 1: **Overall framework of the proposed surface diffusion model.** (A) The training procedure of the conditional DDPM model, where CT denotes cortical thickness, SI denotes shape index and G/S seg. denotes Gyral/Sulcal segmentation. (B) The sampling process for each test subject to generate abnormal score for analysis (Note: all images are actual data and actual generated samples).

neighborhood of each vertex. The network in (Zhao et al., 2021) utilizes neighborhood averaging for pooling and up-pooling,which we empirically find to introduce grid artifacts into the generated samples. Therefore, we employ a different pooling and up pooling method. Utilizing the natural structure of the icosahedron, the pooling for ith order is defined as only keeping vertices in the (i-1)-th order icosahedron and up pooling is the zero padding for vertices added from i-th to (i+1)-th order. Fig. 2 shows the structure of the network and illustrations of the operations. The network has a standard UNet structure with 2 ResBlocks in each level. Each ResBlock has an additional time embedding input, from Sinusoidal embedding layer + MLP layer, same as in (Ho et al., 2020) for denoisnig at each time step. For memory efficiency, the attention layer is only included in the last two levels.

## 2.4. Sampling for Normative Modeling

The core idea for our normative modeling is to use sampled feature maps to measure deviation scores as opposed to registered real data. Through procedures described in previous sections, the model will generate N samples per test subject conditioned on original cortical feature maps with 500 steps of added noise, which is determined empirically, individual anatomical segmentation, sex and age. This step aims to reconstruct disease feature maps to be pseudo-healthy ones while still maintain the same anatomical structure. For each

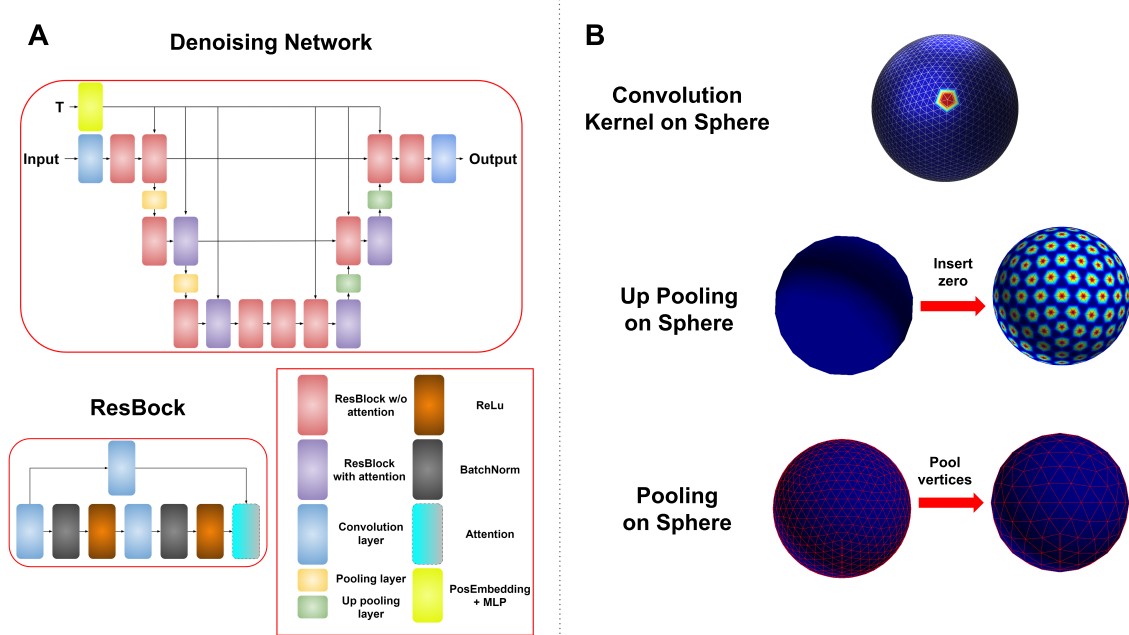

Figure 2: **Denoising network structure and operations in spherical domain:** (A) A UNet structure with ResBlock built out of spherical convolution, pooling, up-pooling and attention layers. (B) Illustration of spherical operations.

Region of Interest (ROI) defined by FreeSurfer output file *aparc.annot*, an abnormal score is computed as in equation 4.

$$Z_i = \frac{x_i - mean([x_{(i,1)}...x_{(i,N)}])}{std_j([x_{(i,1)}...x_{(i,N)}])} \tag{4}$$

For a test subject, $Z_i$ is the abnormal score for the i-th ROI. $x_i$ is the mean feature value of the test subject in the i-th ROI. $x_{(i,j)}$ denotes the mean feature value for the i-th ROI of the j-th sample. The abnormal score measures the deviation from normal in each ROI. Additionally, the abnormal scores of 34 ROIs are used as feature in a standard SVM for 10-fold cross validation of CN vs MCI and CN vs AD classification.

## 3. Experiments and Results

### 3.1. Preprocessing and Implementation

Two public datasets are used in the experiments. 584 subjects are selected from Human Connectome Project (HCP) dataset(Van Essen et al., 2012). 9:1 train test split is applied for unconditional task. 646 subjects are selected from the Alzheimer's Disease Neuroimaging Initiative(ADNI) dataset(Mueller et al., 2005), including 482 CN, 82 MCI, and 82 AD patients. 400 CN subjects are used as training set and all others as test set. All the T1 MRI images are processed through FreeSurfer 6.0 (Dale et al., 1999) to extract the

| type/FID score | CT(front)↓ | CT(back)↓ | Curv(front)↓ | Curv(back)↓ | Sulc(front)↓ | Sulc(back)↓ |
|---|---|---|---|---|---|---|
| VAE | 4.2268 | 5.2528 | 1.4641 | 2.3283 | 0.5640 | **0.3503** |
| SiT Diffusion | 0.4179 | 0.5556 | 0.3526 | 0.3693 | 0.4169 | 0.9080 |
| Our Model | **0.3453** | **0.3103** | **0.2308** | **0.2670** | **0.2351** | 0.4725 |
| Test data | 0.0091 | 0.0080 | 0.0077 | 0.0064 | 0.0037 | 0.0023 |

Table 1: **FID score for each feature map:**. The generated feature maps are embedded to 2D images. The FID score is computed between generated sample or test data and train data.

cortical surfaces and cortical thickness (CT) map, curvature(curv) and sulcal depth(sulc). Surfaces are registered by FreeSurfer in the spherical domain. The Desikan-Killiany Atlas in FreeSurfer is used for ROI parcellation. The shape index (SI) map and gyral/sulcal segmentation mask are obtained following (Shi et al., 2008). All feature maps and masks are resampled to a 6th order icosahedron(40962 vertices) using the *mris_surf2surf* command in FreeSurfer. All feature maps are standardized to 0 mean and std 1. Sex label is set as female:0 and male:1. Age label is scaled to [0,1] range by dividing by 100. For computational costs, the experiments are only conducted on the left hemisphere.

The input to the network is the concatenation of feature maps including CT, SI, Curv, Sulc, and the segmentation mask based on the task. In the unconditional task, we use CT,Curv and Sulc as input. In normative modeling, we use CT,SI,age,sex, and segmentation mask. All feature maps are 40962 length vector. Hidden dimension of each network level is 128,256 and 512. The max timesteps of DDPM is set at 1000. The model is trained with ADAM (Kingma, 2014) as optimizer, cosine annealing(Loshchilov and Hutter, 2016) as scheduler and a starting learning rate of 1e-5 for a total of 1000 epochs, about 24 hours. The network is implemented using Pytorch and trained on a NVIDIA A5000 GPU.

### 3.2. Generating Ability through FID Score

To show the superior generative power of our model, we performed the unconditional generative task on the HCP dataset and compared the performance in terms of the FID score(Heusel et al., 2017) with two other generative models, Variational Autoencoder(VAE) (Kingma, 2013) and surface transformer based diffusion model(Xie et al., 2024). We use the same structure as our backbone in the VAE by removing the skip connections and adding fully connected layer between encoder and decoder. We also included the FID score between test data and training data as a reference. Since FID is designed for 2D images, we embed surface feature maps on fsaverage as snapshots in front and back sagittal views using the jet colormap in Plotly.(Inc., 2015). We use the 192 dimension embedding for FID. Each model generates 200 sets of feature maps and are compared to the training data by FID score. Examples are shown in appendix A. Table 1 shows that, with the exception of sulc back view, our model achieved the best FID score among generative models and is closest to the real data. This result demonstrats that our backbone in DDPM is able to generate new feature maps closer to the real data distribution.

| Model type | SI SSIM↑ | SI MSE(mm) ↓ | CT SSIM↑ | CT MSE(mm) ↓ |
|---|---|---|---|---|
| DDPM | 0.4911 ± 0.0165 | 0.1461 ± 0.0064 | 0.3978 ± 0.0168 | 0.4329 ± 0.0174 |
| DDPM + mask | **0.6167 ± 0.0127** | **0.1011 ± 0.0046** | **0.4903 ± 0.0224** | **0.3894 ± 0.0253** |

Table 2: **Ablation study results:** For both CT and SI, the anatomical condition improves SSIM and MSE.

### 3.3. Ablation Study for Conditional DDPM

We performed ablation study for conditional DDPM to demonstrate improvement. Two models are trained on the 400 CN subjects: unconditional DDPM and DDPM with gyral/sulcal segmentation. All test data are first blurred with 500 time steps of noise, then denoised for sampling. Both models include sex and age conditions. From the 82 test CN subjects, the mean Structural Similarity(SSIM) and mean squared error(MSE) between samples and real data are shown in Table 2. From the ablation study results, we show that the conditioning can indeed improve the sample quality and produce better aligned feature maps.

### 3.4. Normative Assessment on ADNI dataset

To evaluate our model's performance on reducing heterogeneity from anatomical mismatch, we compare our model to spherically registered real data using FreeSurfer (Dale et al., 1999) to compare normative modeling performance using registered data and generated data. The conditional DDPM model is trained on the 400 template CN subjects. After training, for each CN, MCI and AD subject in the test set, 10 samples are generated as the DDPM reference set. To ensure a fair comparison, a template reference set was constructed by selecting 10 subjects from the 400 training CN subjects whose ages are closest to that of the test subject. Abnormality scores for each subject per ROI are computed using both reference sets, following equation 4. The scores were computed using only cortical thickness as an accepted biomarker of brain atrophy in AD.

Qualitative comparison between the real and generated data is illustrated in Fig. 3. All feature maps are resampled to the inflated fsaverage surface for visualization. This figure demonstrates that our model, trained on CN subjects, is capable of estimating the normal feature distribution based on AD subjects' individual anatomical structure, particularly in the temporal region, which strongly correlates to AD pathology. Fig. 4 is a box plot of the mean abnormality scores across cortical ROIs. The statistical difference between CN and MCI, as well as AD, are quantified using ttest p-values for both reference sets. Based on the p-values, our model exhibits increased power in differentiating CN vs MCI and CN vs AD.

Additionally, we also conduct classification experiments for CN vs MCI and CN vs AD. The abnormal scores are computed for 34 ROIs for all test subjects, which are formatted as length 34 vectors. These vectors are then used as features for classification in a standard SVM classifier. To validate the results, we perform 10-fold cross validation and the accuracy,

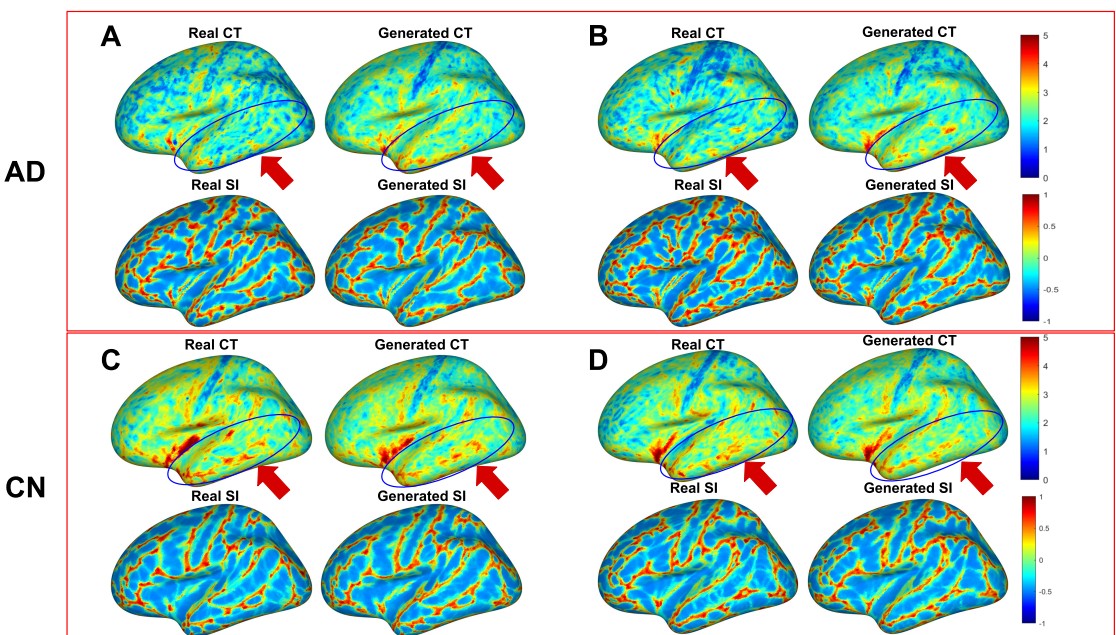

Figure 3: **Comparison between real and generated feature maps for CN and AD subjects** In the figure, CT denotes cortical thickness in unit of millimeter(mm). SI denotes shape index. A, B are 2 AD subjects and C, D are 2 CN subjects. The blue circles highlight the temporal region which highly correlates with AD. A,B demonstrate that our model can infer normal feature distribution for AD subjects. B,C verify that generated feature maps are similar to real ones for CN subject which is expected

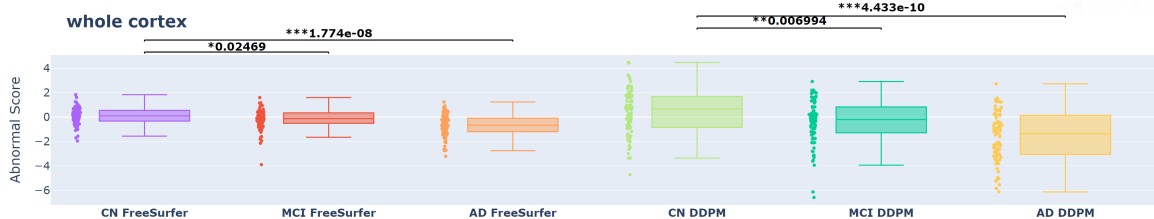

Figure 4: **Mean abnormal score across whole cortex** The figure shows the distribution comparison between mean abnormal score of the whole cortex per subject for template and DDPM reference sets

precision and recall are shown in Table 3. In both CN vs AD and CN vs MCI, our abnormal score performs better than the template's with closest matched age.

| CN vs AD | | | | CN vs MCI | | | |
|---|---|---|---|---|---|---|---|
| Score Type | Accuracy | Precision | Recall | Score Type | Accuracy | Precision | Recall |
| Template | 0.6882 | 0.6482 | 0.7049 | Template | 0.5850 | 0.5850 | 0.5676 |
| DDPM | **0.7128** | **0.7182** | **0.7091** | DDPM | **0.6214** | **0.6554** | **0.6221** |

Table 3: **Classification of CN vs MCI and CN vs AD.** The table shows the accuracy, precision and recall of 10 fold cross validation using template and DDPM reference sets' abnormal score per ROI as feature for SVM.

## 4. Conclusion

In this paper, we proposed a framework for DDPM model on the spherical domain, conditioned on the anatomical segmentation, sex and age to generate anatomically aligned feature maps. The ablation study and normative tests have shown that our model can generate reliable feature maps on the cortical surface and perform better than registered reference set in AD normative modeling. We will freely distribute our source codes and trained models to the research community and enable researchers to utilize our model for other generative tasks on surfaces beyond normative analyses.

## 5. Acknowledgments

This work was supported by the National Institute of Health (NIH) under grants RF1AG077578, RF1AG064584, R01EB022744, RF1AG084072, U19AG078109, and P30AG066530.

Data used in preparing this article were obtained from the ADNI database (adni. loni.usc.edu). As such, many investigators within the ADNI contributed to the design and implementation of ADNI and/or provided data but did not participate in analysis or writing of this report. A complete list of ADNI investigators: http://adni.loni.usc. edu/wp-content/uploads/how_to_apply/ADNI_Acknowledgement_List.pdf.

Data were provided [in part] by the Human Connectome Project, WU-Minn Consortium (Principal Investigators: David Van Essen and Kamil Ugurbil; 1U54MH091657) funded by the 16 NIH Institutes and Centers that support the NIH Blueprint for Neuroscience Research; and by the McDonnell Center for Systems Neuroscience at Washington University.

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

## Appendix A. Example images for FID Score Computation

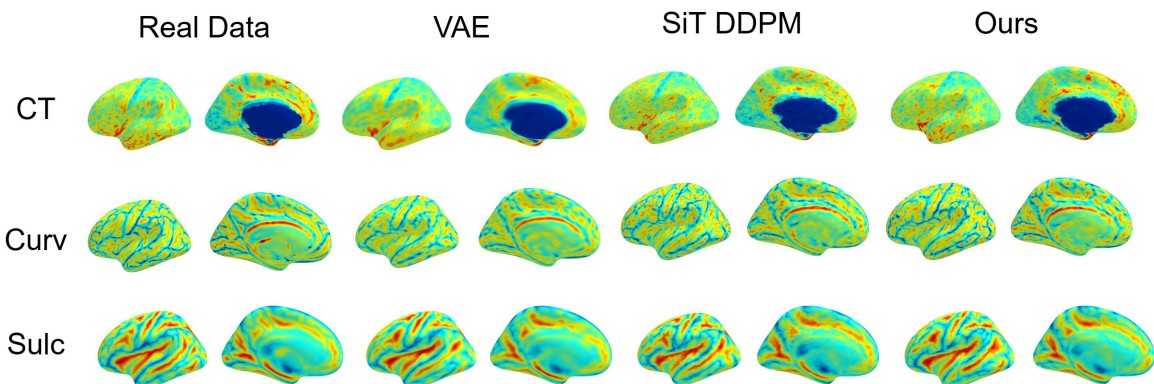

Figure 5: **Example feature maps embedded into 2D images:** Each row shows the saggital front and back view of each type of feature, cortical thickness(CT), curvature(curv) and sulcal depth(sulc). Each column denotes the generative model for the feature maps.

