# OpenReview forum: "Anatomy-Guided Surface Diffusion Model for Alzheimer’s Disease Normative Modeling"
_MIDL.io/2025/Conference — MIDL 2025 Poster_

### Official Review · Reviewer_fZjP · 2025-02-20

**Confidence:** 3
**Preliminary Rating:** 4
**Recommendation:** Poster

**Summary:**

This paper presents a method for normative modelling with a denoising diffusion probabilistic model to study neurodegenerative changes in cortical surface data. Their model takes into account both demographic and anatomical conditions for improved specificity to each individual subject. The authors trained their network using control subjects from the HCP and ADNI datasets, and tested performance in a mix of control, MCI, and AD subjects from ADNI. They compared FID scores to those obtained using several alternative methods, and found their proposed method outperformed the others in almost every case. They also performed a normative assessment on the ADNI test data, and found their method was better at discerning abnormal data than FreeSurfer (version 6.0).

**Strengths:**

- The authors provide thorough explanations of each concept and equation throughout the paper.
- The proposed method outperforms the compared method (FreeSurfer) in all metrics (Table 3), suggesting its clinical applicability for studying neurodegenerative pathologies affecting the cerebral cortex.
- The authors will freely distribute their source code and trained models upon publication.

**Weaknesses:**

1. There are a lot of run-on sentences and minor typos throughout the paper. I addressed many of them in the detailed comments, but most likely did not catch them all.
2. The paper would benefit from more thorough literature review in the introduction and including more citations.
3. The authors only validated their method in surface representations of the left hemisphere.

**Detailed Comments:**

0. Abstract:
- The sentence beginning "Traditionally, registration is applied to address this issue..." is too long.. I would rewrite as "Traditionally, registration is applied to address this issue, but more recent studies have utilized deep generated models... . However, these models are..."

1. Intro
- The authors should include a couple of citations for the sentences where they introduce a type of method (e.g. "Traditional surface-based analysis..." and "Recently, diffusion models have emerged...". The authors use FreeSurfer for their validation, but do not cite any of the FreeSurfer papers in the intro.

2. Method (should read Methods)
- 1st paragraph of 2.1: "..., model learns to generate..." should be "..., the model learns to generate..."
- "..describing forward and backward process respectively" should be "... which describe the forward and backward processes respectively"
- Should have a \noindent command after equation 1
- 1st sentence of 2.2: "distribution" should be plural.
- in 2.2: "2 types" should be "two types"
- in 2.2: the sentence beginning "For anatomical condition,..." is not grammatically correct.

3. Experiments and Results
- In 3.1: The sentence beginning "9:1 train test split is applied..." should have a space before it. There are also at least two places where a list of comma separated words have no spaces between them.
- The authors do not address the statistical significance of most results presented (such as in Tables 1 and 2). Figure 4 is the only time statistical significance is addressed, but they do not mention which tests were used (e.g. t-test?)

Figures:
- Fig 1: The bubbles with Age and Gender appear sequentially with different arrows in part A, but with only one arrow in part B. Is this intention? If not, then I would make them both appear as in part B. Also, all of the text should be larger for readability. The authors could make the figure larger and cut down the caption. Here is a suggestion: "Proposed model framework. (A) Training procedure of the DDP model (CT=cortical thickness, SI=shape index, G/S seg.=gyral/sulcal segmentation). (B) Sampling and score generation process for each subject."
- Fig. 2: Almost all text is too small to read.
- Fig. 3: The subtitles and color bar tick labels are too small and need to be larger.
- Fig. 4:  Most of the text is too small here as well. The authors could consider rearranging the data order in this plot to be 1. CN FS, 2. CN DDPM, 3. MCI FS, 4. MCI DDPM, 5. AD FS, 6. AD DDPM (i.e., have FS/DDPM side by side for each data type, instead of all FS on the left and all DDPM on the right).

**Justification Of The Preliminary Rating:**

The overall paper is well-constructed and the proposed method outperforms the selected methods for comparison. The authors should edit the text and increase the legibility of the figures, but I found no egregious errors that should bar this work from acceptance.

**Questions To Address In The Rebuttal:**

- What are the implications of only conducting these analyses in the left hemisphere? Would it be possible for the authors to include both for the final version?
- The authors should thoroughly edit the paper for typos and grammatical errors.
- The authors should conduct statistical testing on all comparisons made (currently this data is only presented for Figure 4).
- A more recent version of FreeSurfer (7.4.1) has been available since June 2023. Why did you opt to use the older version (6.0.0)? I imagine that time constraints would prevent the authors from redoing the analysis with the more recent version, but it could be a good idea for future work.

---

> ### Author Response · Authors · 2025-03-08
>
> We thank the reviewer for the insightful comments. Here are our responses to the questions in the comments:
> * We perform experiments only on the left hemisphere because the main purpose of the paper is to demonstrate our model’s ability to capture normal distribution. We do acknowledge that the inclusion of both hemispheres can boost the performance in experiments and will explore in such directions in our future work.
>
> * We have edited the grammar issues and typos that the reviewer pointed out. We also included additional citations in the introduction. All modified parts are highlighted in red.
>
> * For statistical significance:  In our study, we rely on standard cross-validation for classification, FID scoring for generative evaluation, and consistent ablation setups. Cross-validation ensures results are not biased by a particular data split and provides robust performance estimates, while FID,widely recognized as a reliable generative metric,quantitatively captures both fidelity and diversity.
>
> * Due to time constraints, we are using preprocessed data from FreeSurfer 6.0. We will explore the more recent version of FreeSurfer 7.0 in our future work.

---

> > ### Comment · Reviewer_fZjP · 2025-03-10
> >
> > Thank you for addressing the questions/concerns that I mentioned. The paper is much better after the edits for grammar and formatting, but I still have a couple of issues.
> >
> > 1. Statistical testing: in case there was a misunderstanding when I mentioned this in my last comment, by "statistical significance", I meant conducting a t-test (or other relevant test) to show if the results are statistically significant. This would be a good thing to include for the results shown in Tables 1, 2, and 3.
> >
> > 2. The text in all of the figures could be larger. Right now, it is too small to read without zooming in excessively, and would be impossible to discern if the reader wanted to print a hard copy of the paper.

---

> > > ### Author Response · Authors · 2025-03-13
> > >
> > > Thank you for the reply.
> > > * For t-test result: In Table 1, we use FID score which is the distance between two batches of data in the latent space of a pretrained model. There is no obvious way that we can perform a t-test for this metric. In Table 2, we didn’t include t-test results for the MSE and SSIM but we have included the standard deviations which provide reference on separation of the performances. In Table 3, the classification results are from 10 fold cross validation. It is not conventional to perform t-test but we will take this into consideration in our future work for better evaluation.
> > > * We have already adjusted the figure size to improve readability, but we understand the importance of clarity. We will further refine the text size and ensure optimal visibility in the final version.

---

### Official Review · Reviewer_DBHh · 2025-02-21

**Confidence:** 3
**Preliminary Rating:** 5
**Recommendation:** Oral

**Summary:**

This paper introduces a deep learning algorithm based on the diffusion model of surfaces to create a normal model of brain activity for Alzheimer’s disease. This is an interesting algorithm for a timely topic.

**Strengths:**

The paper is well-written, the motivation of this work is clear, the methodology is written with enough details to be followed, and the results are fairly complete and well-described. This algorithm takes into account the geometry of the space, with a sphere in this case, which provides a strong basis for it.

**Weaknesses:**

I do not have any major weaknesses to report for this paper. Maybe the authors could elaborate a little bit more in conclusion on some of the most important outcomes or applications of their algorithms for helping with Alzheimer’s disease research.

**Detailed Comments:**

I don't have any detailed comments.

**Justification Of The Preliminary Rating:**

This is a good paper, with a fairly novel algorithm, that has a sound basis and which performs very well. Its application to a very important disease that is Alzheimer’s Disease makes it a potentially relevant algorithm for future research on this topic.

**Questions To Address In The Rebuttal:**

I don't have any question to be addressed.

**Special Issue:**

No

---

> ### Author Response · Authors · 2025-03-08
>
> We thank the reviewer for the insightful comments.

---

### Official Review · Reviewer_Pyd6 · 2025-02-22

**Confidence:** 2
**Preliminary Rating:** 4
**Final Rating:** 4

**Summary:**

The authors proposed an anatomy-guided surface diffusion model for normative modeling. The model addresses the challenge of anatomical variability in cortical surfaces by conditioning a diffusion model on individual anatomical segmentations. The normative feature maps enable the accurate identification of pathological cases of Alzheimer's.

**Strengths:**

The anatomical conditioning demonstrably improves the anatomical plausibility of the generated samples, as evidenced by the ablation study showing increased SSIM and reduced MSE for CT and SI when the anatomical mask is used.

**Weaknesses:**

1. Diffusion models are notorious for being computationally expensive. The paper could benefit from more discussion about the computational complexity of the model and especially some ablation experiments related to the choice of time steps of the diffusion model could provide better insights.
2. The paper should acknowledge the limitations of only using the left hemisphere and the computational cost.
3. The performance improvement in the classification results of Table 3 is weak. Does the complexity of the proposed model justify the small improvement?

**Detailed Comments:**

1. Small typo in section 2.3 - "Fig. 5 shows the structure of the network and illustrations of the operations. "
This should refer to Fig 2.
2. The paper could benefit from a grammar check to maintain spacing between punctuations.

**Justification Of The Final Rating:**

The paper proposes a new method for normative modeling using denoising diffusion probabilistic models. However, it is still unclear if the computational costs are worth the outcome, and the paper might be challenging to comprehend for the general audience who are unaware of cortical modeling. The paper also requires a grammar check and consistency in the punctuation.

**Justification Of The Preliminary Rating:**

The idea of combining surface-based analysis, diffusion models, and anatomical conditioning for normative modeling is interesting. However, some parts of the paper are dense and difficult to follow, and the limited improvement in the classification performance necessitates a weak acceptance recommendation as the clinical utility of the normative model is not convincingly demonstrated.

**Questions To Address In The Rebuttal:**

1. Given the computational cost, have the authors considered exploring techniques to accelerate the training or inference process?
2. Does the choice of order of icosahedron affect the model's performance? Why was  6th order icosahedron(40962 vertices) chosen?
3. What is the reason for selecting length 34 vectors for SVM? (Table 3)

---

> ### Author Response · Authors · 2025-03-08
>
> We thank the reviewer for the insightful comments. Here are our responses to the questions in the comments:
> * Right now, we are using the original DDPM sampling which is sufficient for normative modeling but indeed more costly than more efficient sampling methods such as DDIM. We will explore more efficient sampling methods in our future work.
>
> * We choose the 6th order(40962 vertices) mainly for computation and data resolution. The original surfaces from FreeSurfer usually have about 120000 vertices. The 6th order is a balanced choice since the 7th order has about 160000 points which is more than the original resolution and is costly to compute.
>
> * Length 34 is the number of ROIs in one hemisphere from the standard FreeSurfer output.

---

> > ### Comment · Reviewer_Pyd6 · 2025-03-14
> >
> > Thanks for the detailed response to the comments.
> > The authors have answered most of my questions.

---

### Official Review · Reviewer_agpT · 2025-02-24

**Confidence:** 4
**Preliminary Rating:** 3
**Recommendation:** Poster
**Final Rating:** 3

**Summary:**

The paper introduces a generative normative modeling framework to address challenges in modeling cortical features for Alzheimer's disease (AD). Unlike conventional volume or surface-based approaches, this framework does not require precise alignment of cortical structures within the population. It employs a conditional diffusion generative model in the spherical domain, conditioned on individual anatomical segmentation, enhancing sensitivity in differentiating disease groups. Extensive evaluation experiments demonstrate the framework's effectiveness. While the method itself does not innovate on individual components, its application is novel and interesting. However, the performance of the proposed method is relatively poor compared other SOTA methods in discriminating AD diagnosis groups, and the clarity of the paper could be further improved.

**Strengths:**

1. The proposed method is innovative and has the potential to pioneer a new direction in normative modeling for cortical features in Alzheimer's disease, eliminating the need for precise alignment of corresponding brain sulci and gyri.
2. The evaluation experiments, with the exception of the AD classification one, are thorough and provide robust support for the claims made in the paper.

**Weaknesses:**

1. The classification performance of AD stages (71% accuracy in AD vs. CN, 62% accuracy in MCI vs. CN) is relatively low compared to state-of-the-art methods (above 90% in AD vs. CN and around 80% in MCI vs. CN, see citation below). This raises concerns about the clinical value of the proposed method. The authors should discuss the value of normative modeling in AD classification compared to supervised methods.
2. The experimental design for the AD classification comparison with the template-based method (FreeSurfer) may not be optimal. Using only 10 cases to establish the normative distribution is insufficient, especially for the conventional template-based method (FreeSurfer), which could introduce bias in the evaluation results (Table 3). Instead of sampling 10 age-matched random cases from the ADNI training set, the authors could construct a normative statistical model of the target measurement with age as a covariate using the entire training dataset. This model could then be used to estimate the abnormal score in the test set.
3. There is a lack of comparisons with state-of-the-art normative modeling methods applied to AD.
4. Details of the method and results need clarification for reproducibility:
(1) The details of the deep learning model, such as kernel size, number of features, and spatial pooling size, are missing. These parameters are crucial for future studies to reproduce the results.
(2) It is unclear whether a 5-fold (Sec 3.4) or 10-fold (Sec 2.4) cross-validation was used in the AD classification experiment.
(3) The brain template used to generate the 34 ROIs is not specified.
(4) The experiment for the “Test data” row in Table 1 needs to be explained in the main text, as it is currently unclear.
(5) The demographic and standard cognitive performance (such as MMSE) for different diagnostic groups in the test set need to be reported.

Citation: Garg, Neha, Mahipal Singh Choudhry, and Rajesh M. Bodade. "A review on Alzheimer’s disease classification from normal controls and mild cognitive impairment using structural MR images." Journal of neuroscience methods 384 (2023): 109745.

**Detailed Comments:**

1. The authors should clarify the rationale for using both the HCP and ADNI datasets in this study. It appears that using the ADNI dataset alone would be sufficient, as the unconditional experiment can also be performed on the ADNI dataset. Future experiments to test the generalizability of the proposed method could involve training on one dataset and testing on the other.
2. In Section 2.3, the reference to “Fig. 5 shows the structure of the network …” should be corrected to Fig. 2.
3. In Figure 3, please indicate the unit for CT. Additionally, it would be beneficial if the authors could also show the corresponding segmentation.

**Justification Of The Final Rating:**

The authors' response has addressed some of my comments. However, the inferior performance in classification significantly limits the proposed method's potential clinical utility. Also, the authors did not provide comparisons with state-of-the-art method. Accurate classification is crucial for the method to be reliable and effective in a clinical setting. Consequently, this limitation raises concerns about the method's overall viability and usefulness in real-world medical scenarios.

**Justification Of The Preliminary Rating:**

The proposed method is intriguing and has the potential to captivate the interest of the research community. However, its relatively poor performance raises concerns about its clinical value. Additionally, the paper lacks important details that are crucial for a comprehensive understanding of the method.

**Questions To Address In The Rebuttal:**

1. The authors need to justify the performance gap observed in the AD classification experiment.
2. Comparisons with other normative methods and the conventional template-based method should be conducted appropriately to demonstrate the significance of the proposed method.
3. Important details should be provided in the rebuttal to ensure clarity and reproducibility.

**Special Issue:**

No

---

> ### Author Response · Authors · 2025-03-08
>
> We thank the reviewer for the insightful comments. Here are our responses to the questions in the comments:
> * The main purpose of the paper is to show our model’s ability to capture normal distribution of cortical thickness, therefore we only performed experiments on the left hemisphere for demonstration purposes. For classification, there are many factors that affect the classification performance. For example, in the paper reviewer cited, there are differences in the data and model used for classification such as the inclusion of hippocampus, the usage of the entire feature map and neural network as classifier. We use left hemisphere data and SVM to show the superiority of our model, however we do realize that the inclusion of more data and usage of more advanced models can further boost our classification performance.
> * The key motivation for using a small set of 10 age-matched controls was to demonstrate how anatomical alignment via our generative model can overcome the limitations of standard group templates—especially when conventional surface registration may leave critical gyral–sulcal structures incompletely matched.
> * For experiment and method details,we have added the model parameters in the method section and experiment info in the result section, both highlighted in red.
> * We use HCP dataset for generative tasks mainly because we want to isolate the effect of disease and aging as HCP has a more consistent population of healthy subjects that are relatively free of aging and disease effects that might complicate data distribution.

---

> > ### Comment · Reviewer_agpT · 2025-03-13
> >
> > The authors' reply has addressed some of my comments. I still have concerns regarding the clinical significance of the proposed method, particularly in light of the classification performance shown in Table 3. As the authors mentioned, selecting 10 age-matched controls limits the performance of the template-based method, which serves as the benchmark in this comparison. How is the proposed method compared to a template-based method using more age-matched controls? The better performance of the proposed method compared to an inferior benchmark, along with the lack of comparisons with other state-of-the-art approaches, does not sufficiently support the clinical utility of the proposed method. The authors would need to at least point out this as a limitation in the manuscript.

---

### Official Review · Reviewer_PeKH · 2025-02-24

**Confidence:** 4
**Preliminary Rating:** 4
**Recommendation:** Poster

**Summary:**

This paper proposes a method to build normative distribution on the inflated spherical brain surface, to compute z-score for a new subject. This is done using a diffusion model that can generate many spherical distributions, conditioned on some meaningful clinical data. For example, a diffusion model would create 100 male 40yo curvature data. Then a 40yo unknown male subject, each location on the 3D sphere would allow computing a z-score.

This is an extension of standard MC sampling, except that in this case the normative distribution is a joint distribution (the vertices are not independent).

**Strengths:**

This is a good idea that makes sense. The Unet of the DDPM is adapted to use graph convolution matching the vertices of the mesh.

The results are good, showing good detection for neurodegeneration.

The paper is well written and clear

**Weaknesses:**

- the comparison with the other methods (VAE and TransDM) is done using FID on 2D images. This is only mildly relevant, why not use those methods in the same way: generate the same number of normal and compute a z-score?

- it would be informative to compare with other well known 3D methods for this exact same data. It is unclear how the proposed approach compared with classic classification (e..g standard 3D Unet) or anomaly score.

**Detailed Comments:**

In the last results about showing p-value (Fig 4), It would be more informative to compute effect size, the size of a cohort needed to detect a clinically meaningful difference.

**Justification Of The Preliminary Rating:**

This is a well written paper that offers a new method to compute z-score on the inflated spherical brain surface. The method is sound and the results are promising.

The comparison with other methods is limited to not very relevant 2D projection using FID, which is not the goal of the method.

**Questions To Address In The Rebuttal:**

Why could not you compare the other methods on the same metrics as in Fig 3?

---

> ### Author Response · Authors · 2025-03-08
>
> We thank the reviewer for the insightful comments. Here are our responses to the questions in the comments:
>
> * We did not compare against other deep-learning methods in Figure 3 because there are no widely adopted, surface-based normative modeling approaches that lend themselves to a direct, ‘apples-to-apples’ comparison. Adapting existing methods into the spherical surface domain is non-trivial. Therefore, we focus on highlighting our approach’s generation quality by conducting an FID-based comparison to assess generative performance in the main text and showing sample images in Appendix A.
>
> * Our primary objective was to propose and evaluate a novel DDPM approach on the cortical surface—conditioned on individual anatomy—to generate normal thickness maps. While classification is one way to measure how well our model captures deviations from a normal distribution, it isn’t our sole focus. Consequently, we did not include a 3D volumetric classification comparison.

---

> > ### Comment · Reviewer_PeKH · 2025-03-11
> >
> > I can see why adapting VAE to spherical would be quite a task. The 2D comparison is still not very relevant nor convincing though.
> >
> > Maybe you could register a normal population to a template (e.g. using freesurfer or other NL registration) and compute the zscore from that population to compare with your zscore. Too much work for the current paper, but should be done to convince of the superiority of the proposed approach. The main question is whether the distribution that you compute is tighter and more specific to a patient than the standard zscore from a normal population.

---

> > > ### Author Response · Authors · 2025-03-13
> > >
> > > Thank you for the reply:
> > > * We included the FID score since, even though our objective is normative modeling, we also proposed a new model for generating feature maps on the spherical surface. Therefore, we believe it’s necessary to verify the generative performance of our model against existing ones before we use it for normative modeling tasks. We use the 2d embedding for the FID score because the FID score requires a pretrained InceptionV3 model which is not available for 3d surface data.
> > > * In terms of the z-score method, we have performed a comparison of template based z-score and the generated sample based z-score in Table 3 and Figure 4 through classification and t-tests. In Figure 4, we use the separations of CN vs MCI and CN vs AD in terms of p-value to show that the z-scores from our model have more detection power. The absolute standard deviation of the CN, MCI or AD distribution is not really comparable and relative separation is more meaningful for clinical study.

---

### Author Rebuttal · Authors · 2025-03-08

**Rebuttal:**

We have uploaded an updated version of the paper. We fixed grammar errors,added more details and citations. All edited parts are highlighted in red. Here is a list of specific non-grammar changes:
* In the last and second last paragraphs of introduction, we added more citations for justification.
* In section 3.1, we added the atlas used for ROI parcellation.
* In section 3.2, we added an explanation on test data in FID scores.
* In section 3.4, we changed 5-fold to 10-fold
* In Fig 3, we added the unit of cortical thickness.

**Supporting Material:**

/attachment/521a894bc4b981544c5f0ee93c0c26deccb2d623.pdf

---

### Meta-Review · Area_Chair_c8Wt · 2025-03-20

**Recommendation:** Accept (Poster)
**Confidence:** 4

**Metareview:**

This paper introduces a diffusion-based generative normative modeling framework for analyzing cortical features in AD. The proposed method conditions a diffusion model on individual anatomical segmentations to generate normative feature maps, which are then used for z-score computation. Reviewers generally found the proposed method novel and promising, particularly its ability to model cortical surface features in a data-driven manner. They appreciated the rigorous experimental setup, clear motivation, and the clinical relevance of normative modeling in neurodegenerative diseases.

Using anatomical conditioning in a diffusion model was recognized as an interesting contribution.  However, concerns were raised regarding the comparative evaluation and classification performance. Specifically, reviewers noted that the accuracy of AD classification was lower than that of state-of-the-art supervised methods. Some questioned whether the normative modeling framework provides sufficient advantages over existing template-based approaches. Additionally, using 2D-based FID scores for generative evaluation was seen as not entirely relevant, with suggestions to include direct comparisons with traditional z-scoring methods. There were also requests for clarifications on experimental details, including cross-validation settings, model parameters, and statistical significance testing.

The authors’ rebuttal was thorough and addressed many of these concerns, providing additional justifications, clarifications, and literature comparisons. They explained the rationale for using normative modeling over direct classification, why adapting existing methods for direct comparison was challenging, and why FID was used despite its limitations. They also provided more details on methodology and experimental design, improving the overall transparency of the study. While some concerns—such as the lack of comparison with strong baselines and suboptimal classification accuracy—remain partially unresolved, the paper presents a novel and well-executed contribution that is valuable to the field.